# PSSD-Transformer: Powerful Sparse Spike-Driven Transformer for Image Semantic Segmentation

A

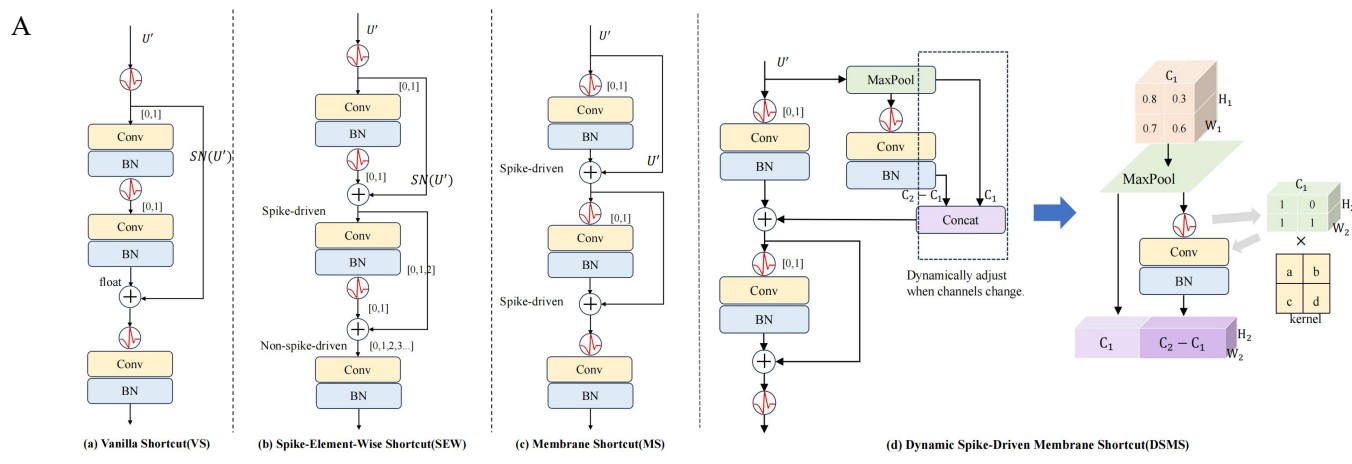

B

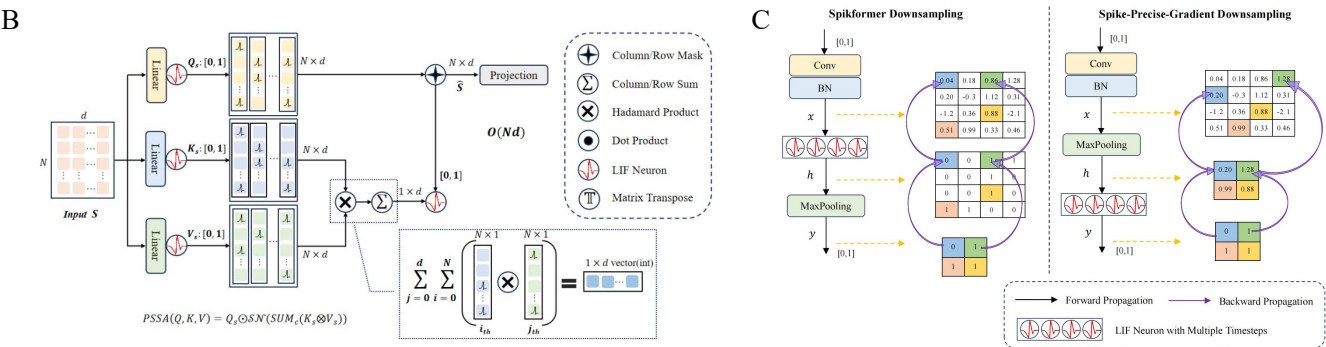

**Figure 1: Presentation of the modules proposed in this paper. (A) shows the details of DSMS and its comparison with other residual connections, (B) illustrates the details of the proposed additive attention mechanism, PSSA, and (C) compares the proposed SPGD with the vanilla method. Detailed introductions of each module are provided in Sec.4.**

## ABSTRACT

Spiking Neural Networks (SNNs) have indeed shown remarkable promise in the field of computer vision, emerging as a low-energy alternative to traditional Artificial Neural Networks (ANNs). However, SNNs also face several challenges: i) Existing SNNs are not purely additive and involve a substantial amount of floating-point computations, which contradicts the original design intention of adapting to neuromorphic chips; ii) The incorrect positioning of convolutional and pooling layers relative to spiking layers leads to reduced accuracy; iii) Leaky Integrate-and-Fire (LIF) neurons have limited capability in representing local information, which is disadvantageous for downstream visual tasks like semantic segmentation.

To address the challenges in SNNs, i) we introduce Pure Sparse Self Attention (PSSA) and Dynamic Spiking Membrane Shortcut (DSMS), combining them to tackle the issue of floating-point computations; ii) the Spiking Precise Gradient downsampling (SPGdown) method is proposed for accurate gradient transmission; iii) the Group-LIF neuron concept is introduced to ensure LIF neurons' capability in representing local information both horizontally and vertically, enhancing their applicability in semantic segmentation tasks. Ultimately, these three solutions are integrated into the Powerful Sparse-Spike-Driven Transformer (PSSD-Transformer), effectively handling semantic segmentation tasks and addressing the challenges inherent in SNNs. The experimental results demonstrate that our model outperforms previous results on standard

**Unpublished working draft. Not for distribution.**

classification datasets and also shows commendable performance on semantic segmentation datasets. Up to this point, PSSD is the first model in the SNN field to perform semantic segmentation on large datasets. The code will be made publicly available after the paper is accepted for publication.

## CCS CONCEPTS

• **Networks → Network on chip**.

## KEYWORDS

Spiking Neuron Networks, Semantic Segmentation

## 1 INTRODUCTION

**Introduction of SNNs.** Spiking Neural Networks (SNNs) are a type of neural network that more closely mimic biological neural networks. Unlike traditional Artificial Neural Networks (ANNs) that use continuous values, SNNs operate using discrete events or "spikes." These spikes are binary events that occur at points in time, resembling the way biological neurons transmit information. SNNs are known for their energy efficiency and are considered more biologically realistic. They have potential applications in areas where low power consumption[15, 27, 33, 39] is crucial, such as edge computing and neuromorphic hardware[3, 9, 11, 35].

**Introduction of semantic segmentation.** Semantic segmentation[16, 29, 41, 46] is a computer vision task where the goal is to assign a label to each pixel in an image such that pixels with the same label share certain characteristics. This process essentially divides the image into meaningful parts with semantic interpretations, like identifying objects, boundaries, or regions. Semantic segmentation is widely used in various applications, including autonomous vehicles[5, 19, 23, 45], medical imaging[14, 38], and scene understanding[8, 42]. The challenge lies in accurately classifying each pixel while maintaining the integrity of object shapes and contextual relationships within the image.

**Why combining the spike mechanisms with segmantic segmentation tasks?** Semantic segmentation, a task of assigning accurate semantic categories to each pixel in an image, plays a crucial role in image understanding by identifying and dividing different semantic regions in a scene. This vision-intensive computation task has significant application potential in various real-world scenarios like autonomous driving, medical surgery robots, and wearable devices. In these contexts, neuromorphic chips with sparse spike-driven properties and low-power computational rules are ideal due to their efficiency and suitability for these applications. The deployment of SNNs on neuromorphic chips is considered a key approach for implementing vision models, such as image segmentation models, marking a promising direction in the future development of artificial intelligence.

**How to combine the spike mechanisms with semantic segmentation tasks?** To answer this question, we consider two perspectives: the contradiction between semantic segmentation's computational density and the low-power nature of spike-driven computation, and the need for spatial consistency and semantic accuracy in semantic segmentation. i) To address the core issue of the contradiction between the computationally dense nature of

semantic segmentation and the low-power spike-driven computation rules, on one hand, the introduction of **Pure Sparse Self Attention (PSSA)** allows the semantic segmentation model to selectively focus on the most important areas for the task, ignoring irrelevant features and fully utilizing the sparsity of model parameters. On the other hand, the introduction of **Dynamic Spiking Membrane Shortcut (DSMS)** reduces the information loss caused by the use of sparse spike-driven attention, achieving sparse spike-driven computation, significantly reducing energy consumption and easing the application of image semantic segmentation models on neuromorphic chips. ii) Regarding the need for spatial consistency and semantic accuracy in semantic segmentation, the spiking Transformer inherits the global semantic information modeling capabilities of the Transformer for long-distance dependencies. The core issue then is how to enhance spatial consistency and focus on local information. On one hand, the introduction of a locality-based **Spiking Precise Gradient downsampling (SPG-down)** mechanism preserves gradients at the points of maximum feature information during backpropagation, and uses DSMS to better preserve spatial information. On the other hand, by migrating **Group-LIF** neurons from the temporal domain to the spatial domain in the feedforward network, considering the neighborhood relationships of pixels in different directions based on feature grouping, local feature contexts are obtained, addressing the problem of weak local information relevance caused by long distances, thereby enhancing the spatial consistency and correlation of features.

Our contributions can be summarized as follows:

- **Group-LIF for Local Information Control**: Developing Group-LIF neurons to enhance control over local information representation.
- **PSSA + DSMS for Non-floating Point Operations**: Implementing Pure Sparse Self Attention (PSSA) combined with Dynamic Spiking Membrane Shortcut (DSMS) to achieve spike-based processing without floating-point computations.
- **SPG-down for Precise Gradient Transmission**: Introducing Spiking Precise Gradient downsampling (SPG-down) to ensure more accurate gradient propagation.
- **PSSD-Transformer Architecture**: Proposing the Powerful Sparse-Spike-Driven Transformer (PSSD-Transformer) architecture and applying it to image classification and semantic segmentation tasks, marking the first application of a spiking Transformer architecture to large-scale semantic segmentation datasets.

## 2 RELATED WORKS

### 2.1 SNNs

*2.1.1 Training Strategy.* There are two approaches to obtaining SNN networks. i) ANN2SNN [4, 6, 24], where an ANN is initially trained and then transformed into an SNN, which involves replacing the ReLU activation layer in a trained ANN network with a spiking neuron and adjusting certain hyper-parameters to achieve a highly accurate SNN. However, this method is characterized by long conversion time-steps as well as constraints associated with the original ANN design. ii) Directly training. In recent research, researchers predominantly employ Spike Time Dependent Plasticity (STDP) for training to simulate synaptic information transmission,

with more recent studies [30, 44, 48] often utilizing surrogate functions for gradient propagation.

*2.1.2 Applications.* Implementing above techniques has led to significant results in various fields. For instance, object detection can be achieved through Spiking-Yolo [20] and ems-yolo [37]. In terms of large language models, SpikingGPT [49] and SpikingBert [2] are preferable, and for generative models, SpikingGAN [22] is the way to go. SpikingGCN [50] and SpikingGAT [40] are recommended for Graph Neural Networks. Furthermore, neuromorphic chips such as TrueNorth [28], Loihi [10], and Tianjic [31] are now available, making it increasingly likely that SNNs will be widely used in the near future.

## 2.2 Semantic Segmentation

*2.2.1 Traditional semantic segmentation.* This section primarily introduces semantic segmentation algorithms based on deep neural networks, categorizing them into four main representative types: i) Fully Convolutional Network (FCN) [26], by replacing fully connected layers with convolutional layers and adopting an encoder-decoder architecture, facilitates the extraction of deep discriminative features for subsequent instance localization and segmentation tasks. However, the computational demands on high-resolution images may result in challenges such as insufficient memory and slower processing speed, while exhibiting a relatively weaker capability in handling detailed boundary information. ii) SegNet [1] employs a pooling index mechanism for upsampling its lower-resolution input feature maps, aiming to accurately reconstruct object contours during the decoding process. However, when dealing with large-sized images, it may encounter significant computational and memory overhead. iii) DeepLab [7] utilizes Dilated Convolution to expand the receptive field, enhancing segmentation performance by capturing contextual information more effectively. iv) strategies based on Visual Transformer (ViT) [13]: SETR [47] achieves pixel-level segmentation through a multi-level feature aggregation module; Segmenter [36] obtains class labels and predicts segmentation masks through a point-wise linear decoder or a Mask Transformer decoder; Segformer [43] employs a simple yet efficient MLP decoder to aggregate information from different layers, cleverly combining local attention and global attention to present powerful representations; PVT overcomes the low-resolution output features of a single scale by introducing a progressively contracting pyramid network backbone.

*2.2.2 Sparse semantic segmentation.* Unlike traditional counterpart, sparse networks commonly employ lightweight architectures to minimize parameter count and computational load, a critical consideration for deploying semantic segmentation models on resource-constrained devices. ERFNet [32] is a lightweight network designed for real-time semantic segmentation, employing residual connections and decomposed convolutions to reduce computational load, suitable for embedded systems. SGCPNet [17] proposes a strategy for spatial detail-guided context propagation, achieving sparsity by avoiding the maintenance of high-resolution features throughout the network. Other strategies like sparse self-attention [18], which decomposing the dense affinity matrix into the product of two sparse matrices. Researchers also employ SNNs to achieve sparsity.

SpikeCalib [25] utilizes FCN for ANN to SNN conversion in semantic segmentation, introducing Burst-Spikes neuron model and proposing Lateral Inhibition Pooling (LIPooling) to address errors from max-pooling during the conversion process. SpikeSEG [21] is the first algorithm for semantic segmentation that directly trains SNNs using spike events with the STDP method, but it is effective primarily on simpler datasets. Our work addresses the gap in SNNs' capability for semantic segmentation in complex scenarios.

## 3 PRELIMINARIES

In this section, we first introduce Leaky Integrate-and-Fire (LIF) neurons, widely applied within the SNN domain. Following this, we explore traditional attention mechanisms and those already present in SNNs. Then, we introduce existing residual connections in SNNs, analyzing their advantages and limitations. Finally, we discuss the mainstream downsampling operations in SNNs and analyze their shortcomings.

### 3.1 LIF

In SNNs, spike neurons control the release of spikes based on a threshold. In this paper, we use LIF [12] neurons, which work in the following way:

$$U[t] = V[t-1] + \frac{1}{k_\tau}\left(X[t] - (V[t-1] - V_{reset})\right) \quad (1)$$

$$S[t] = \mathcal{H}(U[t] - V_{th}) \quad (2)$$

$$V[t] = U[t]\,(1 - S[t]) + V_{reset}S[t] \quad (3)$$

where $k_\tau$, $V_{th}$, and $V_{reset}$ represent the decay factor, firing threshold, and reset membrane potential, respectively, which are pre-set to default values. The notation $X[t]$ refers to the input at time step $t$, while $U[t]$ denotes the membrane potential. The function $\mathcal{H}(\cdot)$ represents the Heaviside step function. The spike output, denoted by $S[t]$, is calculated based on the membrane potential and the threshold. Additionally, $V[t]$ and $V[t-1]$ signify the temporal output at time t.

### 3.2 Vanilla Self-Attention

*3.2.1 Self-Attention in Transformer.* In the Transformer model, self-attention mechanism is employed to handle relationships between different positions in the input sequence. The self-attention mechanism in Transformer involves three key steps: computing attention weights, weighted sum, and linear mapping. Given an input sequence $X = (x_1, x_2, ..., x_n)$ where each $x_i \in \mathbf{R}^{d_x}$ is an element of the input sequence, three sets of representations $Q, K, V$ (query, key, value) are obtained through linear mappings:

$$Q = XW_Q, \quad K = XW_k, \quad V = XW_V \quad (4)$$

here, $W_Q, W_K, W_V$ are learnable weight matrices.

Next, attention weights are computed. The attention weight between positions $i$ and $j$ is calculated using dot product:

$$\alpha_{ij} = Softmax(\frac{Q \cdot K^T}{\sqrt{d_k}}) \quad (5)$$

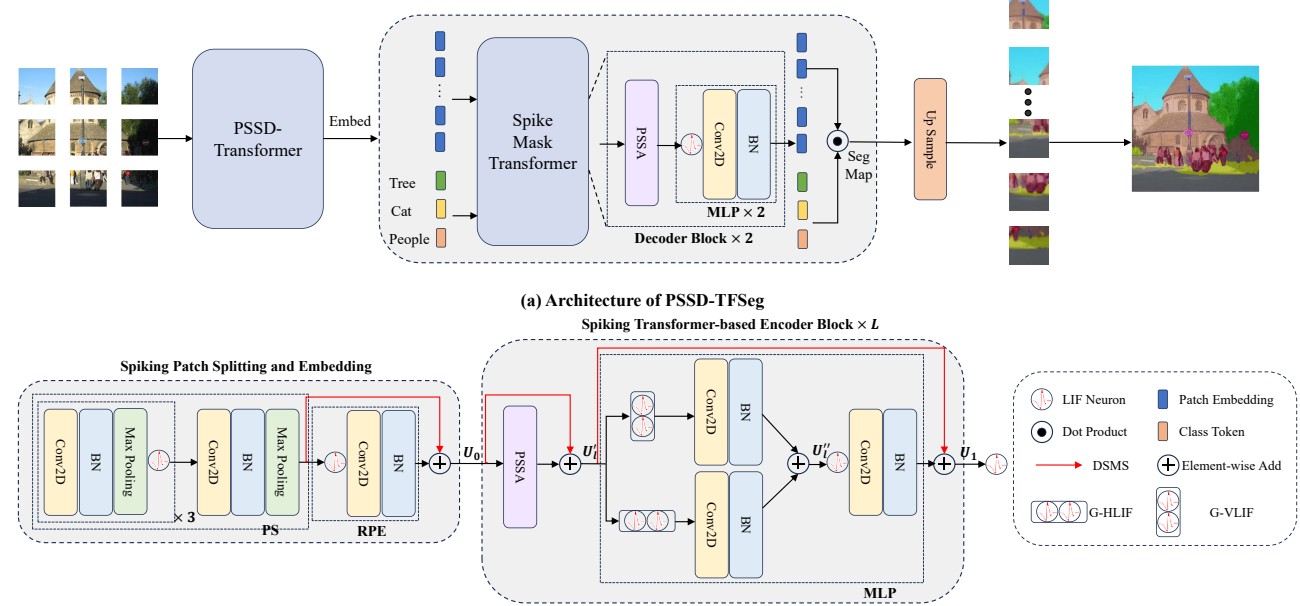

**(a) Architecture of PSSD-TFSeg**

**(b) Details of PSSD-Transformer**

**Figure 2: Illustration of the overall architecture. Initially, the PSSD-Transformer backbone network undergoes pre-training, outputting scores for each category through a linear classification head. The Spike Mask Transformer consists of a PSSA layer and two MLP modules. These components respectively yield a decoded semantic segmentation feature matrix and a category feature embedding. By performing an inner product of these two feature matrices, the model generates K mask sequences for semantic segmentation, where K represents the number of semantic categories. Finally, a bilinear interpolation upsampling is applied to these sequences to match the original image size, resulting in the semantic segmentation output.**

Each output element, $z_i$, is computed as weighted sum of a linearly transformed input elements:

$$z_i = \sum_{j=1}^{n} \alpha_{ij} V \qquad (6)$$

*3.2.2 Spiking Self-Attention in Spikformer.* In Spiking Self-Attention (SSA) mechanism, the input undergoes linear transformations and neuron operations to be transformed into spike-form queries (Q), keys (K), and values (V) containing only 0 and 1. The computed attention map has natural non-negativity, disregarding irrelevant features. Therefore, using floating-point representations for Q, K, V, and the softmax function is redundant for modeling such spike sequences.

$$Q = SN_Q[BN(XW_Q)] \qquad (7)$$
$$K = SN_K[BN(XW_K)] \qquad (8)$$
$$V = SN_V[BN(XW_V)] \qquad (9)$$
$$SSA(Q, K, V) = SN(\frac{QK^T V}{s}) \qquad (10)$$

where $SN_Q, SN_K, SN_V$ and $W_Q, W_K, W_V$ represent Spiking Neurons and weights of the Q, K, V branches, respectively. $Q, K, V \in \mathbf{R}^{T \times N \times d}$, in this context, T denotes the time-step, N denotes the length of the sequence, and d denotes the dimension. BN denotes the Batch Normalization layer, and X is the input of the SSA. s represents the scale factor.

## 3.3 Shortcut

The Membrane Shortcut (MS) has been demonstrated to satisfy the dynamic isometry theory. It adds a shortcut connection to the membrane potential of each spiking neuron, achieving an identity mapping. This ensures that the tensor of spike transmission always contains binary pulse signals.

$$X^L = F^{res}(X^{L-1}) + F^{short}(X^{L-1}) \qquad (11)$$
$$F^{res}(X^{L-1}) = ConvBN(SN(X^{L-1})) \qquad (12)$$

where $X^L$ represents the output of the layer L, $F^{res}(\cdot)$ denotes the residual path, $F^{short}(\cdot)$ denotes the shortcut path, $ConvBN(\cdot)$ represents the operation combining Convolution and Batch Normalization.

However, the MS residual connection method still has some limitations. When changing the channel number to match the feature channel number on the residual path and the shortcut path, it may easily overlook non-sparse and non-spike-driven operations in this part. Non-sparse spike operations can lead to significant energy consumption in this module.

## 3.4 Downsampling in Spikformer

Input undergoes a combination of convolution and batch normalization operations (ConvBN), followed by activation of spiking neurons over multiple time steps. Neurons accumulate membrane

potential over time based on the received current, generating a spike signal when the membrane potential exceeds a threshold. Finally, max-pooling is performed for downsampling.

The backward propagation stage of the above process can be expressed by Eq.13-15:

$$\frac{\partial y_{ij}}{\partial h_{uv}} = \begin{cases} 1 & h_{uv} = max(h) \\ 0 & h_{uv} < max(h) \end{cases} \tag{13}$$

$$\frac{\partial h_{uv}}{\partial x_{uv}} = \frac{\partial X^{t,n}}{\partial V^{t,n}} = \frac{1}{\alpha} H(|V_i^{t,n} - V_{th}| \le \frac{\alpha}{2}) \tag{14}$$

$$\frac{\partial L}{\partial x_{uv}} = \sum_{i=0}^{\frac{H}{s}} \sum_{j=0}^{\frac{W}{s}} \frac{\partial L}{\partial y_{ij}} \frac{\partial y_{ij}}{\partial h_{uv}} \frac{\partial h_{uv}}{\partial x_{uv}}$$
$$= \begin{cases} \frac{1}{\alpha} \frac{\partial L}{\partial y_{ij}} H(|V_i^{t,n} - V_{th}|) & h_{uv} = max(h) \\ 0 & h_{uv} < max(h) \end{cases} \tag{15}$$

where $h \in \mathbf{R}^{H \times W}, x \in \mathbf{R}^{H \times W}, y \in \mathbf{R}^{\frac{H}{s} \times \frac{W}{s}}$ denote the output of the SN, ConvBN and Max Pooling layer, respectively, $h_{uv}, x_{uv}, y_{ij}$ denote the represent the feature information at a specific position for feature maps $h, x, y$. $s$ is the pooling step. $L$ represents the loss function. $\alpha$ is a parameter designed to ensure the integration of the gradient equals 1 and determines the steepness of the curve.

## 4 METHOD

In Sec.3, we refine our methods to better address semantic segmentation challenges. In this section, we start with the introduction of Group-LIF, enhancing local information processing in LIF neurons. We then present PSSA, a sparser alternative to SSA, ideal for asynchronous neural chips, with parameter integration into thresholds to minimize float count. DSMS adapts to the dynamic channel variations in semantic segmentation, extracting features from varying feature map dimensions and ensuring MS pulse-driven functionality. Additionally, SPGD is introduced to ensure accurate pulse-driven gradient backpropagation essential in downsampling operations within semantic segmentation tasks.

### 4.1 Group-LIF

Global LIF neurons introduce weak long-range correlations due to the larger time-steps introduced. In semantic segmentation tasks, it is crucial to maintain spatial consistency, including the interaction of contextual information and local details in the image.

To address the aforementioned issues, we proposes Group-LIF, based on a grouping mechanism. It divides the patch sequence of the feature map into several groups and applies Group-LIF neurons to facilitate information interaction within each group. This enables a more compact pattern of local context information exchange.

As shown in Fig.2(b), G-LIF attempts to transmit information from both horizontal and vertical directions, that is, by learning local contexts and spatial information in different orientations on the image through horizontal LIF and vertical LIF for different communication directions of image blocks. Ablation study for G-LIF is presented in Sec.5. Due to space limitations, details will be provided in the supplementary materials.

## 4.2 Pure Sparse Self-Attention

The SSA still involves non-spiking computations, making it challenging to deploy the model on neuromorphic chips. To avoid non-spiking computations and energy-consuming operations like multiplication and addition, it is essential to leverage energy-efficient operations such as logical AND, addition, masking, which require minimal energy consumption.

**Lemma 1.** Transforming matrix dot product using element-wise multiplication and column summation operations incurs almost negligible energy cost.

$$a^T \odot b = \sum_{j=1}^{N} (a_j \otimes b_j) = \sum_{a_j=1} b_j \tag{16}$$

$$A^T \odot B = \sum_{i=1}^{d} \sum_{j=1}^{N} (A_{ij} \otimes B_{ij}) = \sum_{i=1}^{d} \sum_{A_{ij}=1} B_{ij} \tag{17}$$

where $A, B \in \mathbf{R}^{N \times d}$, $a = A_{i1}$ and $b = B_{i2}$, $(\cdot)^T$ denotes the matrix transformation, $\odot$ represents the matrix multiplication, $\otimes$ is the hadamard product.

As shown in Eq.16, when performing the dot product of the transposed binary column vectors $a$ and $b$ with dimensions $N \times 1$, the resulting dimension is $1 \times 1$. This is equivalent to taking the Hadamard product of column vectors $a$ and $b$ to obtain an $N \times 1$ tensor, followed by column-wise summation to get a $1 \times 1$ vector.

As shown in Eq.17, generalizing to the dot product of all columns of matrices A and B, it can be easily transformed using element-wise multiplication and column summation operations. Moreover, the spiking-driven Hadamard product can be considered as a logical bitwise AND operation with almost negligible energy cost, resulting in a $1 \times d$ dimensional matrix.

Extending this theoretical discovery to the dot product calculation of three binary matrices in the spiking self-attention mechanism, this process can be equivalently substituted by spiking-driven computations for the $Q, K, V$ calculation steps in the SSA mechanism.

$$Q_s \odot K_s^T \odot V_s = SN(\sum_{i=1,j=1}^{N,d} Q_s \otimes K_s) \odot V \tag{18}$$
$$= SN(SUM_r(Q_s \otimes K_s)) \oplus V_s$$

where $\oplus$ denotes the mask operation, $Q_s, K_s, V_s \in \mathbf{R}^{N \times d}$, $SUM_r(\cdot)$ is the row summary.

**Lemma 2.** The scaling factor (s) can be incorporated into the threshold $V_{th}$.

$$o^{(t)} = \begin{cases} 1 & \text{if } \frac{x}{s} > V_{th} \\ 0 & \text{otherwise} \end{cases} \tag{19}$$

where $x$ and $o^{(t)}$ are the input and the output of the Spiking Neuron, respectively.

As shown in Eq.19, we update the $V_{th}$ to $V_{th}^{new} = s \cdot V_{th}$.

$$PSSA(Q, k, V) = SN_{V_{th}=V_{th}^{new}}[SUM_r(Q \otimes K) \oplus V] \tag{20}$$

Thus, in PSSA, the computation process is pure sparse, focusing attention on relevant features to reduce computational overhead. It is also spike-driven, introducing nearly energy-free fully binary

spiking computation, significantly reducing power resources and energy costs, aligning with the computational rules of neuromorphic chips.

## 4.3 Dynamic Spike-Driven Membrane Shortcut

DSMS retains the concept of MS connections in the main residual path. In the shortcut path, when the feature channel number needs to change, an additional LIF neuron model is added before ConvBN to transform information into sparse spiking signals.

In the context of semantic segmentation, DSMS adapts to backbone networks that extract features through continuous downsampling using Maxpooling to reduce feature map size and parameters. When the channel number remains the same or decreases, DSMS employs a single-branch concatenation of LIF neurons and ConvBN after Maxpooling. In cases requiring an increase in channel number, DSMS introduces a Concat operation for comprehensive spiking feature reuse. This dynamic control of channels ensures consistency with the residual main branch, allowing DSMS to efficiently extract target features from diverse dimensions and channel numbers. The above process can be described by the following equations.

$$X' = MaxPool(X^{L-1}) \tag{21}$$

$$F^{short}(X^{L-1}) = Concat(X', ConvBN(SN(X'))) \tag{22}$$

where $X'$ is the computation result of the maxpool branch.

The quantity and channel width of DSMS can be dynamically adjusted based on specific tasks. In semantic segmentation tasks, optimizing DSMS improves information flow, and the output flow on the shortcut path conceptually approximates the sum of synaptic inputs to the neuron membrane. This representation of complete spiking features effectively reflects the energy-efficient characteristics of the network, ensuring that all modules in the network are spike-driven.

## 4.4 Spiking Precise-Gradient Downsampling

Unlike the sequence of operations from the order discussed in3.4, the input in SPG-Down undergoes the ConvBN operation to obtain the feature map $x$. Following this, it undergoes max-pooling to obtain a downsized feature map $h$. Finally, the pulse neuron activation produces the feature map $y$.

The backward propagation stage of the above process can be expressed by the Eq.23-25 :

$$\frac{\partial y_{ij}}{\partial x_{uv}} = \begin{cases} x_{uv} & x_{uv} = max(x) \\ 0 & x_{uv} < max(x) \end{cases} \tag{23}$$

$$\frac{\partial y_{ij}}{\partial h_{ij}} = \frac{\partial X^{t,n}}{\partial V^{t,n}} = \frac{1}{\alpha} H(|V_i^{t,n} - V_{th}|) \tag{24}$$

$$\frac{\partial L}{\partial x_{uv}} = \sum_{i=0}^{\frac{H}{s}} \sum_{j=0}^{\frac{W}{s}} \frac{\partial L}{\partial y_{ij}} \frac{\partial y_{ij}}{\partial h_{ij}} \frac{\partial h_{ij}}{\partial x_{uv}}$$

$$= \begin{cases} \frac{1}{\alpha} \frac{\partial L}{\partial y_{ij}} H(|V_i^{t,n} - V_{th}| & x_{uv} = max(x) \\ 0 & x_{uv} < max(x) \end{cases} \tag{25}$$

where $x$ denotes the output of "ConvBN". The output of the LIF neuron is its membrane potential $h \in \mathbb{R}^{H \times W}$, with the maximization goal being the output vector $y \in \mathbb{R}^{S \times S}$, making the gradients $\frac{\partial y}{\partial h}$ and $\frac{\partial L}{\partial y}$ non-zero. The activation function and the non-differentiable membrane potential update equation pose key challenges for gradient propagation. During backpropagation, the non-zero nature of $\frac{\partial L}{\partial y_{ij}}$ and $\frac{\partial y_{ij}}{\partial h_{uv}}$ results in the gradient $\frac{\partial L}{\partial h_{uv}}$ becoming zero, where $x_{uv}$, $h_{uv}$, $y_{ij}$ represent the respective gradients from $\frac{\partial h_{uv}}{\partial x_{uv}}$, then all gradients would vanish during propagation. Each term of the gradient is the product of the derivative of the previous term and the current term, where $\alpha$ is an amplification factor to apply this amplification to all gradient calculations, with $max(h)$ being amplified to a sufficiently large value.

Evidently, the maximum element after max-pooling in the feature map x is selected as the maximum value, representing the position with the most feature information in x. During back propagation, this position retains the gradient, preserving the maximum feature information in each local region before max-pooling (aggregation). Additionally, since max-pooling is applied to feature map x first, the feature map h has a reduced size, saving energy and memory resources.

## 5 EXPERIMENT

In this work, the datasets used include ImageNet, CIFAR-10, and ADE20K. The introductions to these datasets and basic experimental setup are provided in the supplementary materials. In this section, ablation studies are conducted on the various modules proposed in this paper; followed by an analysis of the performance of the proposed PSSD-TFSeg on semantic segmentation datasets. It's worth noting that, as we are the first to apply an SNN model on large-scale semantic segmentation datasets, there are no other models for comparison. The chapter concludes with the presentation of Grad-Cam images and semantic segmentation result images from different layers of the model.

## 5.1 Ablation Study

To investigate the characteristics and effectiveness of several modules proposed in this chapter, including the spiking computing approach (PSSA self-attention, MS and DSMS residual connections), SPG-Down precise gradient downsampling, and neuron mechanisms (Group Horizontal G-HLIF and Group Vertical G-VLIF), a series of ablation experiments were conducted. These experiments further assessed the contributions of the proposed algorithms.

*5.1.1 Feasibility of Sparse spiking.* Experiments were conducted on the ImageNet1K dataset to assess the accuracy and performance of the PSSA self-attention mechanism and MS/DSMS residual connections. This involved ablation studies on SSA and PSSA mechanisms, SEW versus MS/DSMS, and the combined effect of these two modules.

Tab.1 reveals that although replacing SSA with PSSA incurs a slight loss in accuracy, the energy consumption is less than 50% of that of the SSA-based Spikformer. Essentially, the pulse-driven self-attention mechanism utilizes binary self-attention scores to mask non-essential channels in sparse pulse value tensors, converting all multiplications into sparse computational additions. While this

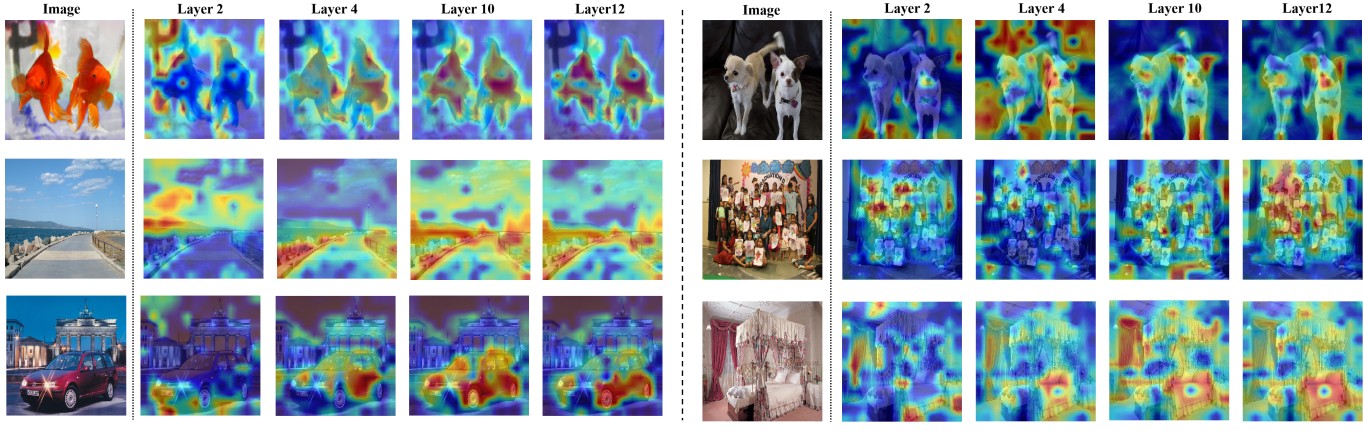

**Figure 3: Grad-CAM visualization of the PSSD-Transformer's mid-layer reveals how it divides images into semantic regions, focusing on key objects within its encoder. This targeted learning enhances class label prediction for pixel segmentation by honing in on significant object areas, improving semantic category understanding.**

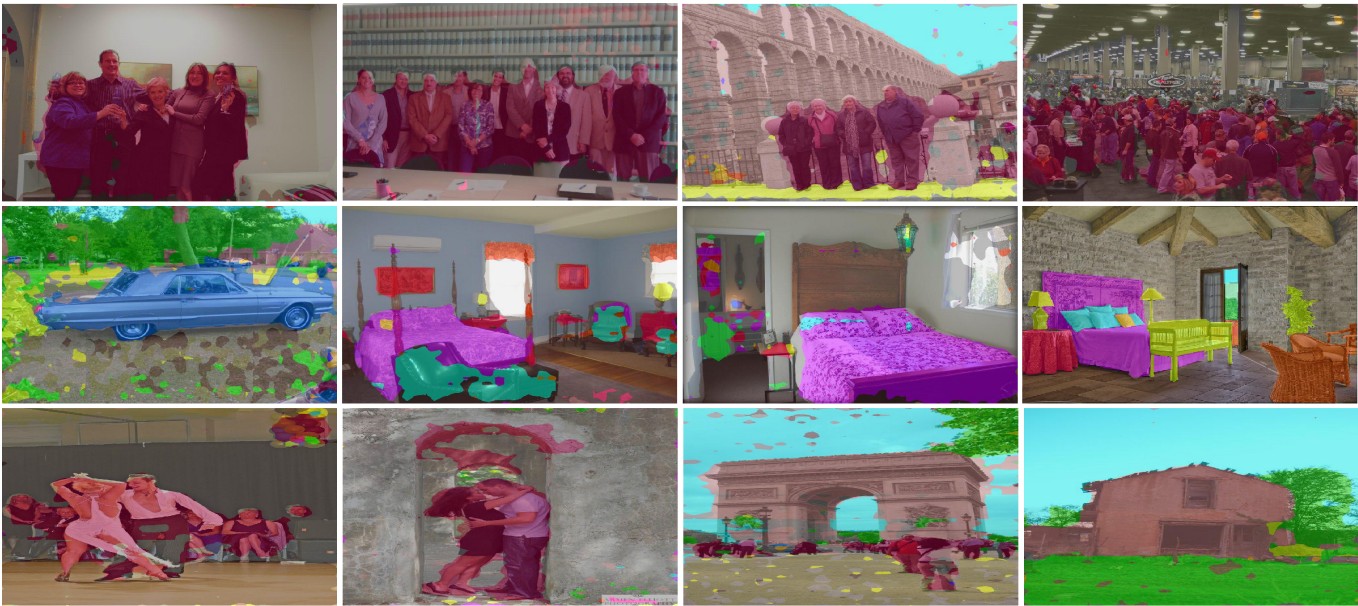

**Figure 4: Showcased in the image are semantic segmentation results from the PSSD-TFSeg model on the ADE20K dataset, featuring an MLP-based segmentation head and MS residual connections in its encoder-decoder structure. The model adeptly handles a variety of settings, effectively focusing on objects requiring segmentation in both crowded indoor scenes and complex outdoor environments.**

leads to a minor accuracy reduction, the PSSA operator consumes almost no energy. This not only achieves sparse pulse driving but also aligns with low-power computing principles, bringing the deployment of neuromorphic chips closer to reality.

*5.1.2 Effectiveness of spatial consistency.* Experiments on image classification using backbone networks on the CIFAR10 dataset. These experiments involved replacing the downsampling methods in Spikformer and PSSD-Transformer with either standard Vanilla Downsample or SPG-Down, maintaining a time step of 4.

As shown in Tab.2, this setting demonstrated the effectiveness and rationality of the optimized precise downsampling method.

## 5.2 Comparative quantitative experiments

In this section, comparative quantitative experiments were conducted using the ADE20K dataset (with a resolution of 512 × 512) and three evaluation metrics. The semantic segmentation backbone network employed was PSSD-Transformer, where residual connections utilized both MS and DSMS. Additionally, the model incorporated the pulse downsampling method SPG-Down and neuron

**Table 1: Experiments of the Feasibility of Sparse spiking (PSSA and DSMS) on Imagenet**

|   | Self-Attention | Shortcut | Power(mJ) | Acc (%) |
|---|---|---|---|---|
| 1 | *SSA | *SEW | 4.96 | 65.1 |
| 2 | SSA | MS | 4.92(-0.04) | 66.9(+1.8) |
| 3 | PSSA | SEW | 2.18(-2.78) | 64.5(-0.6) |
| 4 | PSSA | MS | **2.14(-2.82)** | 66.2(+1.1) |
| 5 | PSSA | DSMS+MS | 2.21(-2.75) | **66.8**(+1.7) |

**Table 2: Ablation Study of Spatial Consistency (SPG-Down and G-LIF) on ADE20K**

|   | Architecture | Downsampling | Neuron | MIoU (%) |
|---|---|---|---|---|
| 1 | Spikformer | Vanilla Down | LIF | 26.7 |
| 2 | Spikformer | SPG-Down | LIF | 26.9(+0.2) |
| 3 | Spikformer | SPG-Down | G-HLIF | 27.4(+0.7) |
| 4 | Spikformer | SPG-Down | G-MLIF | 27.8(+1.1) |

combinations based on G-VLIF and G-HLIF. The semantic segmentation decoders used in these experiments included a pulse linear MLP module (MLP) and a Spike Mask Transformer (SMT).

**Table 3: Comparative quantitative experiments on ADE20K**

|   | Architecture | Decoder | Power(mJ) | MIoU(%) |
|---|---|---|---|---|
| 1 | *Spikformer | MLP | 19.12 | 26.7 |
| 2 | Spikformer | SMT | 20.81(+1.69) | 26.9(+0.2) |
| 3 | PSSD + MS | MLP | **12.61(-6.51)** | 28.6(+1.9) |
| 4 | PSSD + MS | SMT | 14.04(-5.08) | 28.9(+2.2) |
| 5 | PSSD + DSMS | MLP | 13.50(-5.62) | **29.1(+2.4)** |

## 5.3 Visualization

Clearly shown in Fig.3, the Grad-CAM[34] of the PSSD-Transformer effectively segments the image into distinct semantic regions, aligning with the semantic categories of each area. Within each encoder block of the model, the focus is gradually refined to emphasize key object areas in the actual data. This precise targeting significantly bolsters the model's ability to learn various semantic categories, thereby enhancing its capability to more accurately assign class labels to individual pixels in later segmentation tasks.

Fig.4 presents a glimpse of the PSSD-TFSeg model's semantic segmentation capabilities on the ADE20K dataset. This model features an MLP-based semantic segmentation head and MS for residual connections within its encoder-decoder framework. Excelling in a variety of scenes, the model adeptly focuses on key objects for segmentation. This includes intricate indoor settings with numerous people and objects, as well as outdoor landscapes populated with cars, buildings, and animals.

## 6 DISCUSSION

### 6.1 Energy Consumption Estimation

The essence of SNN algorithms is to reconcile the trade-off between precision and power usage. This segment delves into how spike-based models achieve efficiency, utilizing FLOPS and SOPs as performance indicators. SOPs, indicating spikes per second, mirror an SNN's complexity; fewer SOPs mean lower power draw. PSSD-Transformer's efficiency stems from Self-Attention, ConvBN, and MLP modules, which skip computations when inputs are inactive. Energy use for ConvBN and MLP is calculable via operator energy, ANN FLOPS, SFR, and timestep count. The energy consumption of each component is reflected in Tab.1 and Tab.3. Above process can be described by Eq.26-28.

$$E_{\text{Base}} = E_{\text{AC}} \times \text{FL} \times R \times T \tag{26}$$

$$E_{\text{Attn}} = E_{\text{AC}} \times \text{MSFR} \times N_{\text{LIF}} \times T \tag{27}$$

$$E_{\text{sum}} = E_{\text{ConvBn}} + E_{\text{MLP}} + E_{\text{Attn}} \tag{28}$$

Here, FL represents the known FLOPS of the corresponding architecture ANN, reflecting the spiking rate, $E_{AC}$ represents the energy consumption of a single addition operation in the operators. Generally, in ANN models, $E_{MAC}$ = 4.6pJ, $E_{AC}$ = 0.9pJ, T represents the simulation time step, and $N_{LIF}$ represents the number of spiking neurons.

### 6.2 Limitation and Future work

The application of our work on other datasets has not yet been completed, but it has already demonstrated strong performance on the ADE20K dataset. Future work will focus on both depth and breadth: depth refers to deeper and more powerful networks, while breadth refers to more fields and datasets. Another point is that, similar to other works in this field, energy consumption estimation is still in a preliminary phase. Experiments are conducted on GPUs and have not yet been actually deployed on neuromorphic chips. We believe that in the near future, neuromorphic chips will support a wider range of networks and find extensive applications across various fields, shining brightly. By then, PSSD will become even more meaningful.

## 7 CONCLUSION

In this work, we introduce the PSSD-Transformer, a novel approach that effectively marries SNNs with the demanding task of semantic segmentation. By addressing the challenges faced by SNNs in computer vision, such as their intensive floating-point computations, inaccurate layer positioning, and limited local information representation, we have paved the way for the first large-scale application of SNNs in semantic segmentation. Our proposed solutions, including PSSA, DSMS, SPG-down, and Group-LIF neurons, collectively enhance the model's efficiency, accuracy, and applicability. Demonstrating significant improvements on various datasets, the PSSD-Transformer not only surpasses existing models in performance but also highlights the untapped potential of SNNs in energy-efficient, high-precision visual tasks.

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
