# OpenReview forum: "PSSD-Transformer: Powerful Sparse Spike-Driven Transformer for Image Semantic Segmentation"
_acmmm.org/ACMMM/2024/Conference — MM2024 Poster_

### Official Review · Reviewer_4h3c · 2024-05-22

**Rating:** 1
**Confidence:** 4

**Summary:**

This paper proposes Powerful Sparse-Spike-Driven Transformer (PSSD-Transformer), a spiking Transformer for image semantic segmentation. It employs four techniques: Group-LIF neuron, PSSA mechanism, DSMS shortcut, and SPG-down. Ablation experiments demonstrate the effectiveness of the proposed techniques.

**Strengths:**

+ This paper discusses in detail the design of residual shortcuts for spiking neural networks, which is commonly ignored in SNN works.

**Limitations:**

- Lines 118-120 state that the PSSD is the first model in the SNN field to perform semantic segmentation on large datasets. However, Spike-driven Transformer v2 [1] has performed semantic segmentation on the ADE20K dataset and achieved better performance (up to 35.3% MIoU). Therefore, this claim is misleading.
- The proposed PSSA mechanism lacks novelty. It is similar to the SDSA mechanism proposed in Spike-driven Transformer [2]. Please detail the novelty of PSSA compared to SDSA.
- The performance of PSSD lags behind the state-of-the-art SNN model Spike-driven Transformer v2.
- Figure 1 is not well drawn. the three sub-figures A, B, and C are not stylized uniformly. In addition, these sub-figures are not vector graphics, making the text inside difficult to recognize.



[1] Yao, M., Hu, J., Hu, T., Xu, Y., Zhou, Z., Tian, Y., ... & Li, G. (2024). Spike-driven transformer v2: Meta spiking neural network architecture inspiring the design of next-generation neuromorphic chips. In *The Twelfth International Conference on Learning Representations*.

[2] Yao, M., Hu, J., Zhou, Z., Yuan, L., Tian, Y., Xu, B., & Li, G. (2024). Spike-driven transformer. *Advances in Neural Information Processing Systems*, *36*.

**Suitability:**

1

---

### Official Review · Reviewer_sAPs · 2024-05-24

**Rating:** 5
**Confidence:** 3

**Summary:**

Proposed the Pure Sparse Self Attention (PSSA) and Dynamic Spiking Membrane Shortcut (DSMS) to address the issues of floating-point computations and inaccurate layer positioning in SNNs. Introduced the Spiking Precise Gradient downsampling (SPG-down) method to enable accurate gradient backpropagation for downsampling operations in semantic segmentation. Developed the Group-LIF neurons to enhance the local information representation capability of LIF neurons.

**Strengths:**

1. A pulse-based self-attention mechanism (PSSA) and residual connection (MS and DSMS) are proposed, which can significantly reduce energy consumption with only a slight loss of accuracy.12
2. An optimized accurate downsampling method (SPG-Down) is proposed, which can improve the spatial consistency and thus improve the performance of semantic segmentation.

**Limitations:**

1. The formatting of the formulas is inconsistent; some are centered, some are aligned to the left, and others to the right.
2. The clarity of Figure 1(B) is insufficient.

**Suitability:**

3

---

### Official Review · Reviewer_7cfM · 2024-05-26

**Rating:** 4
**Confidence:** 2

**Summary:**

mentation.
To address the challenges in SNNs, i) we introduce Pure Sparse
Self Attention (PSSA) and Dynamic Spiking Membrane Shortcut
(DSMS), combining them to tackle the issue of floating-point computations; ii) the Spiking Precise Gradient downsampling (SPGdown) method is proposed for accurate gradient transmission; iii)
the Group-LIF neuron concept is introduced to ensure LIF neurons’ capability in representing local information both horizontally
and vertically, enhancing their applicability in semantic segmentation tasks. Ultimately, these three solutions are integrated into the
Powerful Sparse-Spike-Driven Transformer (PSSD-Transformer),
effectively handling semantic segmentation tasks and addressing
the challenges inherent in SNNs. The experimental results demonstrate that our model outperforms previous results on standard

**Strengths:**

• Group-LIF for Local Information Control: Developing
Group-LIF neurons to enhance control over local information
representation.
• PSSA + DSMS for Non-floating Point Operations: Implementing Pure Sparse Self Attention (PSSA) combined with
Dynamic Spiking Membrane Shortcut (DSMS) to achieve
spike-based processing without floating-point computations.
• SPG-down for Precise Gradient Transmission: Introducing Spiking Precise Gradient downsampling (SPG-down) to
ensure more accurate gradient propagation.
• PSSD-Transformer Architecture: Proposing the Powerful Sparse-Spike-Driven Transformer (PSSD-Transformer)
architecture and applying it to image classification and semantic segmentation tasks, marking the first application of
a spiking Transformer architecture to large-scale semantic
segmentation datasets.

**Limitations:**

it's hard to train, and how to get snn information

**Suitability:**

2

---

### Official Review · Reviewer_vaLd · 2024-05-28

**Rating:** 3
**Confidence:** 3

**Summary:**

The paper introduces the PSSD-Transformer, a novel model designed to improve the performance of Spiking Neural Networks (SNNs) in the domain of image semantic segmentation. This model addresses several challenges faced by existing SNNs, including the reliance on floating-point computations, poor positioning of convolutional and pooling layers, and the limited local information representation of Leaky Integrate-and-Fire (LIF) neurons. The authors propose three key components: Pure Sparse Self Attention (PSSA), Dynamic Spiking Membrane Shortcut (DSMS), and Spiking Precise Gradient downsampling (SPG-down). These innovations are combined in the PSSD-Transformer architecture, which has demonstrated superior performance on standard classification and semantic segmentation datasets.

**Strengths:**

1) Innovative Components: Introduction of PSSA, DSMS, and SPG-down addresses specific limitations of traditional SNNs, improving overall performance.
2) Experiment: The ablation experiment is well set up.
3) The workload of the paper is substantial.

**Limitations:**

1) Many details in the article require careful adjustment by the author. Here just briefly list some. In terms of formulas: the problem of alignment between (21) and (22), the problem of switching formula (25), and A_{ij=1} in formula (17). There are also lack of symbol descriptions behind the formulas. For example, the R and MSFR symbols in formulas 26-28 need to be explained after the formulas; section references: such as in section 4.4. "Unlike the sequence of operations from the order discussed in3.4, “ this could be written as “Unlike the sequence of operations from the order discussed in sec 3.4,”; citation: Some citations need to be reconfirmed, such as “[2] Malyaban Bal and Abhronil Sengupta. 2023. SpikingBERT: Distilling BERT to Train Spiking Language Models Using Implicit Differentiation. arXiv preprint arXiv:2308.10873 (2023).” But as far as I know, this article has been published at AAAI2024. Please pay attention again to the issue of citation format. At the same time, the article lacks citations, such as "Generally, in ANN models, E_{MAC}= 4.6pJ, E_{AC} = 0.9pJ," this data needs to be cited; Please readjust the abstract format.
2) Lack of comparative in Experiments. In your paper, “ It’s worth noting that, as we are the first to apply an SNN model on largescale semantic segmentation datasets, there are no other models for comparison. ” I hope to see you guys applying your model on small scale semantic segmentation datasets to compare with previous SNN models and show your model’s ability in small datasets.
3) Neuromorphic Hardware. If this paper cannot apply real Hardware experiments, please mention less hardware advantages in the article. Low computational effort is enough. If you really wants to mention the hardward. Please add the specific hardware requirements to achieve the energy efficiency benefits described.
4) Training Efficiency: How does the training time of the PSSD-Transformer compare to traditional ANN-based models for semantic segmentation?
5) Group-LIF: Could you please tell me the specific formula and derivation of Group-LIF?

**Suitability:**

2

---

### Meta-Review · Area_Chair_RKyF · 2024-06-27

**Recommendation:** Accept (Poster)
**Confidence:** 5

**Metareview:**

The paper introduces the PSSD-Transformer, a novel model for improving Spiking Neural Networks (SNNs) in image semantic segmentation. Reviewers have noted the innovation and substantial contributions of components like Pure Sparse Self Attention (PSSA), Dynamic Spiking Membrane Shortcut (DSMS), and Spiking Precise Gradient downsampling (SPG-down). Despite some concerns regarding formula formatting, clarity of illustrations, and comparative analysis, the majority of reviewers have acknowledged the solid empirical results and the paper's significant contribution to the field. Consequently, I recommend accepting this submission.